# Neural Architecture Optimization

[1]**Renqian Luo**[†][*], [2]**Fei Tian**[†], [2]**Tao Qin**, [1]**Enhong Chen**, [2]**Tie-Yan Liu**
[1]University of Science and Technology of China, Hefei, China
[2]Microsoft Research, Beijing, China
[1]lrq@mail.ustc.edu.cn, cheneh@ustc.edu.cn
[2]{fetia, taoqin, tie-yan.liu}@microsoft.com

## Abstract

Automatic neural architecture design has shown its potential in discovering powerful neural network architectures. Existing methods, no matter based on reinforcement learning or evolutionary algorithms (EA), conduct architecture search in a discrete space, which is highly inefficient. In this paper, we propose a simple and efficient method to automatic neural architecture design based on continuous optimization. We call this new approach neural architecture optimization (NAO). There are three key components in our proposed approach: (1) An encoder embeds/maps neural network architectures into a continuous space. (2) A predictor takes the continuous representation of a network as input and predicts its accuracy. (3) A decoder maps a continuous representation of a network back to its architecture. The performance predictor and the encoder enable us to perform gradient based optimization in the continuous space to find the embedding of a new architecture with potentially better accuracy. Such a better embedding is then decoded to a network by the decoder. Experiments show that the architecture discovered by our method is very competitive for image classification task on CIFAR-10 and language modeling task on PTB, outperforming or on par with the best results of previous architecture search methods with a significantly reduction of computational resources. Specifically we obtain $2.11\%$ test set error rate for CIFAR-10 image classification task and $56.0$ test set perplexity of PTB language modeling task. The best discovered architectures on both tasks are successfully transferred to other tasks such as CIFAR-100 and WikiText-2. Furthermore, combined with the recent proposed weight sharing mechanism, we discover powerful architecture on CIFAR-10 (with error rate $3.53\%$) and on PTB (with test set perplexity $56.6$), with very limited computational resources (less than 10 GPU hours) for both tasks.

## 1 Introduction

Automatic design of neural network architecture without human intervention has been the interests of the community from decades ago [12, 22] to very recent [45, 46, 27, 36, 7]. The latest algorithms for automatic architecture design usually fall into two categories: reinforcement learning (RL) [45, 46, 34, 3] based methods and evolutionary algorithm (EA) based methods [40, 32, 36, 27, 35]. In RL based methods, the choice of a component of the architecture is regarded as an action. A sequence of actions defines an architecture of a neural network, whose dev set accuracy is used as the reward. In EA based method, search is performed through mutations and re-combinations of architectural components, where those architectures with better performances will be picked to continue evolution.

It can be easily observed that both RL and EA based methods essentially perform search within the discrete architecture space. This is natural since the choices of neural network architectures are

---

[*]The work was done when the first author was an intern at Microsoft Research Asia.
[†]The first two authors contribute equally to this work.

typically discrete, such as the filter size in CNN and connection topology in RNN cell. However, directly searching the best architecture within discrete space is inefficient given the exponentially growing search space with the number of choices increasing. In this work, we instead propose to *optimize* network architecture by mapping architectures into a continuous vector space (i.e., network embeddings) and conducting optimization in this continuous space via gradient based method. On one hand, similar to the distributed representation of natural language [33, 25], the continuous representation of an architecture is more compact and efficient in representing its topological information; On the other hand, optimizing in a continuous space is much easier than directly searching within discrete space due to better smoothness.

We call this optimization based approach *Neural Architecture Optimization* (NAO), which is briefly shown in Fig. 1. The core of NAO is an encoder model responsible to map a neural network architecture into a continuous representation (the blue arrow in the left part of Fig. 1). On top of the continuous representation we build a regression model to approximate the final performance (e.g., classification accuracy on the dev set) of an architecture (the yellow surface in the middle part of Fig. 1). It is noteworthy here that the regression model is similar to the performance predictor in previous works [4, 26, 10]. What distinguishes our method is how to leverage the performance predictor: different with previous work [26] that uses the performance predictor as a heuristic to select the already generated architectures to speed up searching process, we directly *optimize* the module to obtain the continuous representation of a better network (the green arrow in the middle and bottom part of Fig. 1) by gradient descent. The optimized representation is then leveraged to produce a new neural network architecture that is predicted to perform better. To achieve that, another key module for NAO is designed to act as the decoder recovering the discrete architecture from the continuous representation (the red arrow in the right part of Fig. 1). The decoder is an LSTM model equipped with an attention mechanism that makes the exact recovery easy. The three components (i.e., encoder, performance predictor and decoder) are jointly trained in a multi task setting which is beneficial to continuous representation: the decoder objective of recovering the architecture further improves the quality of the architecture embedding, making it more effective in predicting the performance.

We conduct thorough experiments to verify the effectiveness of NAO, on both image classification and language modeling tasks. Using the same architecture space commonly used in previous works [45, 46, 34, 26], the architecture found via NAO achieves $2.11\%$ test set error rate (with cutout [11]) on CIFAR-10. Furthermore, on PTB dataset we achieve $56.0$ perplexity, also surpassing best performance found via previous methods on neural architecture search. Furthermore, we show that equipped with the recent proposed weight sharing mechanism in ENAS [34] to reduce the large complexity in the parameter space of child models, we can achieve improved efficiency in discovering powerful convolutional and recurrent architectures, e.g., both take less than $10$ hours on 1 GPU.

Our codes and model checkpoints are available at `https://github.com/renqianluo/NAO`.

## 2 Related Work

Recently the design of neural network architectures has largely shifted from leveraging human knowledge to automatic methods, sometimes referred to as Neural Architecture Search (NAS) [40, 45, 46, 26, 34, 6, 36, 35, 27, 7, 6, 21]. As mentioned above, most of these methods are built upon one of the two basic algorithms: reinforcement learning (RL) [45, 46, 7, 3, 34, 8] and evolutionary algorithm (EA) [40, 36, 32, 35, 27]. For example, [45, 46, 34] use policy networks to guide the next-step architecture component. The evolution processes in [36, 27] guide the mutation and recombination process of candidate architectures. Some recent works [17, 18, 26] try to improve the efficiency in architecture search by exploring the search space incrementally and sequentially, typically from shallow to hard. Among them, [26] additionally utilizes a performance predictor to select the promising candidates. Similar performance predictor has been specifically studied in parallel works such as [10, 4]. Although different in terms of searching algorithms, all these works target at improving the quality of discrete decision in the process of searching architectures.

The most recent work parallel to ours is DARTS [28], which relaxes the discrete architecture space to continuous one by mixture model and utilizes gradient based optimization to derive the best architecture. One one hand, both NAO and DARTS conducts continuous optimization via gradient based method; on the other hand, the continuous space in the two works are different: in DARTS it is the mixture weights and in NAO it is the embedding of neural architectures. The difference in

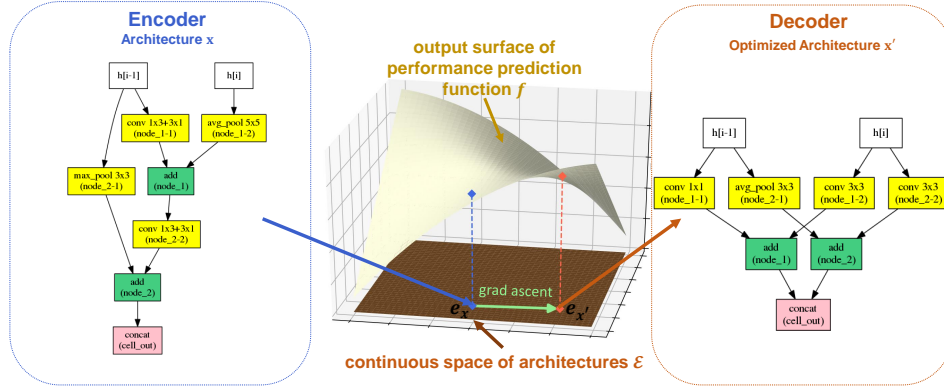

Figure 1: The general framework of NAO. Better viewed in color mode. The original architecture $x$ is mapped to continuous representation $e_x$ via encoder network. Then $e_x$ is optimized into $e_{x'}$ via maximizing the output of performance predictor $f$ using gradient ascent (the green arrow). Afterwards $e_{x'}$ is transformed into a new architecture $x'$ using the decoder network.

optimization space leads to the difference in how to derive the best architecture from continuous space: DARTS simply assumes the best decision (among different choices of architectures) is the $argmax$ of mixture weights while NAO uses a decoder to exactly recover the discrete architecture.

Another line of work with similar motivation to our research is using bayesian optimization (BO) to perform automatic architecture design [37, 21]. Using BO, an architecture's performance is typically modeled as sample from a Gaussian process (GP). The induced posterior of GP, a.k.a. the acquisition function denoted as $a : \mathcal{X} \to R^+$ where $\mathcal{X}$ represents the architecture space, is tractable to minimize. However, the effectiveness of GP heavily relies on the choice of covariance functions $K(x, x')$ which essentially models the similarity between two architectures $x$ and $x'$. One need to pay more efforts in setting good $K(x, x')$ in the context of architecture design, bringing additional manual efforts whereas the performance might still be unsatisfactory [21]. As a comparison, we do not build our method on the complicated GP setup and empirically find that our model which is simpler and more intuitive works much better in practice.

## 3 Approach

We introduce the details of neural architecture optimization (NAO) in this section.

### 3.1 Architecture Space

Firstly we introduce the design space for neural network architectures, denoted as $\mathcal{X}$. For fair comparison with previous NAS algorithms, we adopt the same architecture space commonly used in previous works [45, 46, 34, 26, 36, 35].

**For searching CNN architecture**, we assume that the CNN architecture is hierarchical in that a cell is stacked for a certain number of times (denoted as $N$) to form the final CNN architecture. The goal is to design the topology of the cell. A cell is a convolutional neural network containing $B$ nodes. Each of the nodes contains two branches, with each branch taking the output of one of the former nodes as input and applying an operation to it. The operation set includes 11 operators listed in Appendix. The node adds the outputs of its two branches as its output. The inputs of the cell are the outputs of two previous cells, respectively denoted as node $-2$ and node $-1$. Finally, the outputs of all the nodes that are not used by any other nodes are concatenated to form the final output of the cell. Therefore, for each of the $B$ nodes we need to: 1) decide which two previous nodes are used as the inputs to its two branches; 2) decide the operation to apply to its two branches. We set $B = 5$ in our experiments as in [46, 34, 26, 35].

**For searching RNN architecture**, we use the same architecture space as in [34]. The architecture space is imposed on the topology of an RNN cell, which computes the hidden state $h_t$ using input $i_t$ and previous hidden state $h_{t-1}$. The cell contains $B$ nodes and we have to make two decisions for

each node, similar to that in CNN cell: 1) a previous node as its input; 2) the activation function to apply. For example, if we sample node index 2 and ReLU for node 3, the output of the node will be $o_3 = ReLU(o_2 \cdot W_3^h)$. An exception here is for the first node, where we only decide its activation function $a_1$ and its output is $o_1 = a_1(i_t \cdot W^i + h_{t-1} \cdot W_1^h)$. Note that all $W$ matrices are the weights related with each node. The available activation functions are: tanh, ReLU, identity and sigmoid. Finally, the output of the cell is the average of the output of all the nodes. In our experiments we set $B = 12$ as in [34].

We use a sequence consisting of discrete string tokens to describe a CNN or RNN architecture. Taking the description of CNN cell as an example, each branch of the node is represented via three tokens, including the node index it selected as input, the operation type and operation size. For example, the sequence "node-2 conv 3x3 node1 max-pooling 3x3 " means the two branches of one node respectively takes the output of node $-2$ and node 1 as inputs, and respectively apply $3 \times 3$ convolution and $3 \times 3$ max pooling. For the ease of introduction, we use the same notation $x = \{x_1, \cdots, x_T\}$ to denote such string sequence of an architecture $x$, where $x_t$ is the token at $t$-th position and all architectures $x \in \mathcal{X}$ share the same sequence length denoted as $T$. $T$ is determined via the number of nodes $B$ in each cell in our experiments.

### 3.2 Components of Neural Architecture Optimization

The overall framework of NAO is shown in Fig. 1. To be concrete, there are three major parts that constitute NAO: the *encoder*, the *performance predictor* and the *decoder*.

*Encoder.* The encoder of NAO takes the string sequence describing an architecture as input, and maps it into a continuous space $\mathcal{E}$. Specifically the encoder is denoted as $E : \mathcal{X} \rightarrow \mathcal{E}$. For an architecture $x$, we have its continuous representation (a.k.a. embedding) $e_x = E(x)$. We use a single layer LSTM as the basic model of encoder and the hidden states of the LSTM are used as the continuous representation of $x$. Therefore we have $e_x = \{h_1, h_2, \cdots, h_T\} \in \mathcal{R}^{T \times d}$ where $h_t \in \mathcal{R}^d$ is LSTM hidden state at $t$-th timestep with dimension $d$[3].

*Performance predictor.* The performance predictor $f : \mathcal{E} \rightarrow \mathcal{R}^+$ is another important module accompanied with the encoder. It maps the continuous representation $e_x$ of an architecture $x$ into its performance $s_x$ measured by dev set accuracy. Specifically, $f$ first conducts mean pooling on $e_x = \{h_1, \cdots, h_T\}$ to obtain $\bar{e}_x = \frac{1}{T} \sum_t^T h_t$, and then maps $\bar{e}_x$ to a scalar value using a feed-forward network as the predicted performance. For an architecture $x$ and its performance $s_x$ as training data, the optimization of $f$ aims at minimizing the least-square regression loss $(s_x - f(E(x)))^2$ .

Considering the objective of performance prediction, an important requirement for the encoder is to guarantee the permutation invariance of architecture embedding: for two architectures $x_1$ and $x_2$, if they are symmetric (e.g., $x_2$ is formed via swapping two branches within a node in $x_1$), their embeddings should be close to produce the same performance prediction scores. To achieve that, we adopt a simple data augmentation approach inspired from the data augmentation method in computer vision (e.g., image rotation and flipping): for each $(x_1, s_x)$, we add an additional pair $(x_2, s_x)$ where $x_2$ is symmetrical to $x_1$, and use both pairs (i.e., $(x_1, s_x)$ and $(x_2, s_x)$) to train the encoder and performance predictor. Empirically we found that acting in this way brings non-trivial gain: on CIFAR-10 about 2% improvement when we measure the quality of performance predictor via the pairwise accuracy among all the architectures (and their performances).

*Decoder.* Similar to the decoder in the neural sequence-to-sequence model [38, 9], the decoder in NAO is responsible to decode out the string tokens in $x$, taking $e_x$ as input and in an autoregressive manner. Mathematically the decoder is denoted as $D : \mathcal{E} \rightarrow x$ which decodes the string tokens $x$ from its continuous representation: $x = D(e_x)$. We set $D$ as an LSTM model with the initial hidden state $s_0 = h_T(x)$. Furthermore, attention mechanism [2] is leveraged to make decoding easier, which will output a context vector $ctx_r$ combining all encoder outputs $\{h_t\}_{t=1}^T$ at each timestep $r$. The decoder $D$ then induces a factorized distribution $P_D(x|e_x) = \prod_{r=1}^T P_D(x_r|e_x, x_{<r})$ on $x$, where the distribution on each token $x_r$ is $P_D(x_r|e_x, x_{<r}) = \frac{\exp(W_{x_r}[s_r, ctx_r])}{\sum_{x' \in V_r} \exp(W_{x'}[s_r, ctx_r])}$. Here $W$ is the output embedding matrix for all tokens, $x_{<r}$ represents all the previous tokens before position $r$, $s_r$ is the

LSTM hidden state at $r$-th timestep and $[,]$ means concatenation of two vectors. $V_r$ denotes the space of valid tokens at position $r$ to avoid the possibility of generating invalid architectures.

The training of decoder aims at recovering the architecture $x$ from its continuous representation $e_x = E(x)$. Specifically, we would like to maximize $\log P_D(x|E(x)) = \sum_{r=1}^T \log P_D(x_r|E(x), x_{<r})$. In this work we use the vanilla LSTM model as the decoder and find it works quite well in practice.

### 3.3 Training and Inference

We jointly train the encoder $E$, performance predictor $f$ and decoder $D$ by minimizing the combination of performance prediction loss $L_{pp}$ and structure reconstruction loss $L_{rec}$:

$$L = \lambda L_{pp} + (1-\lambda)L_{rec} = \lambda \sum_{x \in X}(s_x - f(E(x))^2 - (1-\lambda)\sum_{x \in X}\log P_D(x|E(x)), \qquad (1)$$

where $X$ denotes all candidate architectures $x$ (and their symmetrical counterparts) that are evaluated with the performance number $s_x$. $\lambda \in [0,1]$ is the trade-off parameter. Furthermore, the performance prediction loss acts as a regularizer that forces the encoder not optimized into a trivial state to simply copy tokens in the decoder side, which is typically eschewed by adding noise in encoding $x$ by previous works [1, 24].

When both the encoder and decoder are optimized to convergence, the inference process for better architectures is performed in the continuous space $\mathcal{E}$. Specifically, starting from an architecture $x$ with satisfactory performance, we obtain a better continuous representation $e_{x'}$ by moving $e_x = \{h_1, \cdots, h_T\}$, i.e., the embedding of $x$, along the gradient direction induced by $f$:

$$h'_t = h_t + \eta \frac{\partial f}{\partial h_t}, \quad e_{x'} = \{h'_1, \cdots, h'_T\}, \qquad (2)$$

where $\eta$ is the step size. Such optimization step is represented via the green arrow in Fig. 1. $e_{x'}$ corresponds to a new architecture $x'$ which is probably better than $x$ since we have $f(e_{x'}) \geq f(e_x)$, as long as $\eta$ is within a reasonable range (e.g., small enough). Afterwards, we feed $e_{x'}$ into decoder to obtain a new architecture $x'$ assumed to have better performance [4]. We call the original architecture $x$ as 'seed' architecture and iterate such process for several rounds, with each round containing several seed architectures with top performances. The detailed algorithm is shown in Alg. 1.

---

**Algorithm 1** Neural Architecture Optimization

---

**Input**: Initial candidate architectures set $X$ to train NAO model. Initial architectures set to be evaluated denoted as $X_{eval} = X$. Performances of architectures $S = \emptyset$. Number of seed architectures $K$. Step size $\eta$. Number of optimization iterations $L$.
**for** $l = 1, \cdots, L$ **do**
    Train each architecture $x \in X_{eval}$ and evaluate it to obtain the dev set performances $S_{eval} = \{s_x\}, \forall x \in X_{eval}$. Enlarge $S$: $S = S \bigcup S_{eval}$.
    Train encoder $E$, performance predictor $f$ and decoder $D$ by minimizing Eqn.(1), using $X$ and $S$.
    Pick $K$ architectures with top $K$ performances among $X$, forming the set of seed architectures $X_{seed}$.
    For $x \in X_{seed}$, obtain a better representation $e_{x'}$ from $e_{x'}$ using Eqn. (2), based on encoder $E$ and performance predictor $f$. Denote the set of enhanced representations as $E' = \{e_{x'}\}$.
    Decode each $x'$ from $e_{x'}$ using decoder, set $X_{eval}$ as the set of new architectures decoded out: $X_{eval} = \{D(e_{x'}), \forall e_{x'} \in E'\}$. Enlarge $X$ as $X = X \bigcup X_{eval}$.
**end for**
**Output**: The architecture within $X$ with the best performance

---

### 3.4 Combination with Weight Sharing

Recently the weight sharing trick proposed in [34] significantly reduces the computational complexity of neural architecture search. Different with NAO that tries to reduce the huge computational cost brought by the search algorithm, weight sharing aims to ease the huge complexity brought by massive child models via the one-shot model setup [5]. Therefore, the weight sharing method is complementary to NAO and it is possible to obtain better performance by combining NAO and weight

sharing. To verify that, apart from the aforementioned algorithm 1, we try another different setting of NAO by adding the weight sharing method. Specifically we replace the RL controller in ENAS [34] by NAO including encoder, performance predictor and decoder, with the other training pipeline of ENAS unchanged. The results are reported in the next section 4.

## 4   Experiments

In this section, we report the empirical performances of NAO in discovering competitive neural architectures on benchmark datasets of two tasks, the image recognition and the language modeling.

### 4.1   Results on CIFAR-10 Classification

We move some details of model training for CIFAR-10 classification to the Appendix. The architecture encoder of NAO is an LSTM model with token embedding size and hidden state size respectively set as 32 and 96. The hidden states of LSTM are normalized to have unit length, i.e., $h_t = \frac{h_t}{||h_t||_2^2}$, to constitute the embedding of the architecture $x$: $e_x = \{h_1, \cdots, h_T\}$. The performance predictor $f$ is a one layer feed-forward network taking $\frac{1}{T} \sum_{t=1}^{T} h_t$ as input. The decoder is an LSTM model with an attention mechanism and the hidden state size is 96. The normalized hidden states of the encoder LSTM are used to compute the attention. The encoder, performance predictor and decoder of NAO are trained using Adam for 1000 epochs with a learning rate of 0.001. The trade-off parameters in Eqn. (1) is $\lambda = 0.9$. The step size to perform continuous optimization is $\eta = 10$. Similar to previous works [45, 34], for all the architectures in NAO training phase (i.e., in Alg. 1), we set them to be small networks with $B = 5, N = 3, F = 32$ and train them for 25 epochs. After the best cell architecture is found, we build a larger network by stacking such cells 6 times (set $N = 6$), and enlarging the filter size (set $F = 36, F = 64$ and $F = 128$), and train it on the whole training dataset for 600 epochs.

We tried two different settings of the search process for CIFAR-10: *without weight sharing* and *with weight sharing*. For the first setting, we run the evaluation-optimization step in Alg. 1 for three times (i.e., $L = 3$), with initial $X$ set as 600 randomly sampled architectures, $K = 200$, forming $600 + 200 + 200 = 1000$ model architectures evaluated in total. We use 200 V100 GPU cards to complete all the process within 1 day. For the second setting, we exactly follow the setup of ENAS [34] and the search process includes running 150 epochs on 1 V100 GPU for 7 hours.

The detailed results are shown in Table 1, where we demonstrate the performances of best experts designed architectures (in top block), the networks discovered by previous NAS algorithm (in middle block) and by NAO (refer to its detailed architecture in Appendix), which we name as NAONet (in bottom block). We have several observations. (1) NAONet achieves the best test set error rate (2.11) among all architectures. (2) When accompanied with weight sharing (NAO-WS), NAO achieves 3.53% error rate with much less parameters (2.5M) within only 7 hours (0.3 day), which is more efficient than ENAS. (3) Compared with the previous strongest architecture, the *AmoebaNet*, within smaller search space ($\#op = 11$), NAO not only reduces the classification error rate ($3.34 \rightarrow 3.18$), but also needs an order of magnitude less architectures that are evaluated ($20000 \rightarrow 1000$). (3) Compared with PNAS [26], even though the architecture space of NAO is slightly larger, NAO is more efficient ($M = 1000$) and significantly reduces the error rate of PNAS ($3.41 \rightarrow 3.18$).

We furthermore conduct in-depth analysis towards the performances of NAO. In Fig. 2(a) we show the performances of performance predictor and decoder w.r.t. the number of training data, i.e., the number of evaluated architectures $|X|$. Specifically, among 600 randomly sampled architectures, we randomly choose 50 of them (denoted as $X_{test}$) and the corresponding performances (denoted as $S_{test} = \{s_x, \forall x \in X_{test}\}$) as test set. Then we train NAO using the left 550 architectures (with their performances) as training set. We vary the number of training data as $(100, 200, 300, 400, 500, 550)$ and observe the corresponding quality of performance predictor $f$, as well as the decoder $D$. To evaluate $f$, we compute the pairwise accuracy on $X_{test}$ and $S_{test}$ calculated via $f$, i.e., $acc_f = \frac{\sum_{x_1 \in X_{test}, x_2 \in X_{test}} \mathbb{1}_{f(E(x_1)) \geq f(E(x_2))} \mathbb{1}_{s_{x_1} \geq s_{x_2}}}{\sum_{x_1 \in X_{test}, x_2 \in X_{test}} 1}$, where $\mathbb{1}$ is the 0-1 indicator function. To evaluate decoder $D$, we compute the Hamming distance (denoted as $Dist$) between the sequence representation of decoded architecture using $D$ and original architecture to measure their differences. Specifically the measure is $dist_D = \frac{1}{|X_{test}|} \sum_{x \in X_{test}} Dist(D(E(x)), x)$. As shown in Fig. 2(a), the performance predictor is able to achieve satisfactory quality (i.e., $> 78\%$ pairwise accuracy) with only roughly

| Model | B | N | F | #op | Error(%) | #params | M | GPU Days |
|---|---|---|---|---|---|---|---|---|
| DenseNet-BC [19] | | 100 | 40 | / | 3.46 | 25.6M | / | / |
| ResNeXt-29 [41] | | | | / | 3.58 | 68.1M | / | / |
| NASNet-A [45] | 5 | 6 | 32 | 13 | 3.41 | 3.3M | 20000 | 2000 |
| NASNet-B [45] | 5 | 4 | N/A | 13 | 3.73 | 2.6M | 20000 | 2000 |
| NASNet-C [45] | 5 | 4 | N/A | 13 | 3.59 | 3.1M | 20000 | 2000 |
| Hier-EA [27] | 5 | 2 | 64 | 6 | 3.75 | 15.7M | 7000 | 300 |
| AmoebaNet-A [35] | 5 | 6 | 36 | 10 | 3.34 | 3.2M | 20000 | 3150 |
| AmoebaNet-B [35] | 5 | 6 | 36 | 19 | 3.37 | 2.8M | 27000 | 3150 |
| AmoebaNet-B [35] | 5 | 6 | 80 | 19 | 3.04 | 13.7M | 27000 | 3150 |
| AmoebaNet-B [35] | 5 | 6 | 128 | 19 | 2.98 | 34.9M | 27000 | 3150 |
| AmoebaNet-B + Cutout [35] | 5 | 6 | 128 | 19 | 2.13 | 34.9M | 27000 | 3150 |
| PNAS [26] | 5 | 3 | 48 | 8 | 3.41 | 3.2M | 1280 | 225 |
| ENAS [34] | 5 | 5 | 36 | 5 | 3.54 | 4.6M | / | 0.45 |
| Random-WS | 5 | 5 | 36 | 5 | 3.92 | 3.9M | / | 0.25 |
| DARTS + Cutout [28] | 5 | 6 | 36 | 7 | 2.83 | 4.6M | / | 4 |
| NAONet | 5 | 6 | 36 | 11 | 3.18 | 10.6M | 1000 | 200 |
| NAONet | 5 | 6 | 64 | 11 | 2.98 | 28.6M | 1000 | 200 |
| NAONet + Cutout | 5 | 6 | 128 | 11 | 2.11 | 128M | 1000 | 200 |
| NAONet-WS | 5 | 5 | 36 | 5 | 3.53 | 2.5M | / | 0.3 |

Table 1: Performances of different CNN models on CIFAR-10 dataset. 'B' is the number of nodes within a cell introduced in subsection 3.1. 'N' is the number of times the discovered normal cell is unrolled to form the final CNN architecture. 'F' represents the filter size. '#op' is the number of different operation for one branch in the cell, which is an indicator of the scale of architecture space for automatic architecture design algorithm. 'M' is the total number of network architectures that are trained to obtain the claimed performance. '/' denotes that the criteria is meaningless for a particular algorithm. 'NAONet-WS' represents the architecture discovered by NAO and the weight sharing method as described in subsection 3.4. 'Random-WS' represents the random search baseline, conducted in the weight sharing setting of ENAS [34].

500 evaluated architectures. Furthermore, the decoder $D$ is powerful in that it can almost exactly recover the network architecture from its embedding, with averaged Hamming distance between the description strings of two architectures less than $0.5$, which means that on average the difference between the decoded sequence and the original one is less than $0.5$ tokens (totally $60$ tokens).

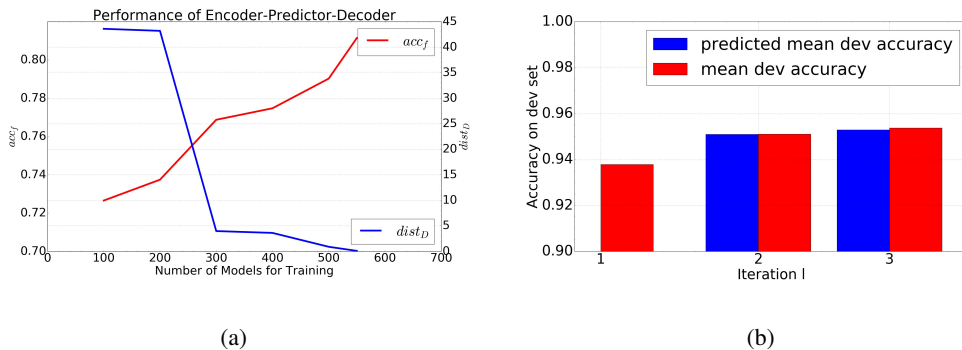

(a)                    (b)

Figure 2: Left: the accuracy $acc_f$ of performance predictor $f$ (red line) and performance $dist_D$ of decoder $D$ (blue line) on the test set, w.r.t. the number of training data (i.e., evaluated architectures). Right: the mean dev set accuracy, together with its predicted value by $f$, of candidate architectures set $X_{eval}$ in each NAO optimization iteration $l = 1, 2, 3$. The architectures are trained for $25$ epochs.

Furthermore, we would like to inspect whether the gradient update in Eqn.(2) really helps to generate better architecture representations that are further decoded to architectures via $D$. In Fig. 2(b) we show the average performances of architectures in $X_{eval}$ discovered via NAO at each optimization iteration. Red bar indicates the mean of real performance values $\frac{1}{|X_{eval}|} \sum_{x \in X_{eval}} s_x$ while blue bar indicates

the mean of predicted value $\frac{1}{|X_{eval}|}\sum_{x\in X_{eval}} f(E(x))$. We can observe that the performances of architectures in $X_{eval}$ generated via NAO gradually increase with each iteration. Furthermore, the performance predictor $f$ produces predictions well aligned with real performance, as is shown via the small gap between the paired red and blue bars.

## 4.2 Transferring the discovered architecture to CIFAR-100

To evaluate the transferability of discovered NAOnet, we apply it to CIFAR-100. We use the best architecture discovered on CIFAR-10 and exactly follow the same training setting. Meanwhile, we evaluate the performances of other automatically discovered neural networks on CIFAR-100 by strictly using the reported architectures in previous NAS papers [35, 34, 26]. All results are listed in Table 2. NAONet gets test error rate of 14.75, better than the previous SOTA obtained with cutout [11](15.20). The results show that our NAONet derived with CIFAR-10 is indeed transferable to more complicated task such as CIFAR-100.

| Model | B | N | F | #op | Error (%) | #params |
|---|---|---|---|---|---|---|
| DenseNet-BC [19] | / | 100 | 40 | / | 17.18 | 25.6M |
| Shake-shake [15] | / | / | / | / | 15.85 | 34.4M |
| Shake-shake + Cutout [11] | / | / | / | / | 15.20 | 34.4M |
| NASNet-A [45] | 5 | 6 | 32 | 13 | 19.70 | 3.3M |
| NASNet-A [45] + Cutout | 5 | 6 | 32 | 13 | 16.58 | 3.3M |
| NASNet-A [45] + Cutout | 5 | 6 | 128 | 13 | 16.03 | 50.9M |
| PNAS [26] | 5 | 3 | 48 | 8 | 19.53 | 3.2M |
| PNAS [26] + Cutout | 5 | 3 | 48 | 8 | 17.63 | 3.2M |
| PNAS [26] + Cutout | 5 | 6 | 128 | 8 | 16.70 | 53.0M |
| ENAS [34] | 5 | 5 | 36 | 5 | 19.43 | 4.6M |
| ENAS [34] + Cutout | 5 | 5 | 36 | 5 | 17.27 | 4.6M |
| ENAS [34] + Cutout | 5 | 5 | 36 | 5 | 16.44 | 52.7M |
| AmoebaNet-B [35] | 5 | 6 | 128 | 19 | 17.66 | 34.9M |
| AmoebaNet-B [35] + Cutout | 5 | 6 | 128 | 19 | 15.80 | 34.9M |
| NAONet + Cutout | 5 | 6 | 36 | 11 | 15.67 | 10.8M |
| NAONet + Cutout | 5 | 6 | 128 | 11 | 14.75 | 128M |

Table 2: Performances of different CNN models on CIFAR-100 dataset. 'NAONet' represents the best architecture discovered by NAO on CIFAR-10.

## 4.3 Results of Language Modeling on PTB

| Models and Techniques | #params | Test Perplexity | GPU Days |
|---|---|---|---|
| Vanilla LSTM [43] | 66M | 78.4 | / |
| LSTM + Zoneout [23] | 66M | 77.4 | / |
| Variational LSTM [14] | 19M | 73.4 | |
| Pointer Sentinel-LSTM [31] | 51M | 70.9 | / |
| Variational LSTM + weight tying [20] | 51M | 68.5 | / |
| Variational Recurrent Highway Network + weight tying [44] | 23M | 65.4 | / |
| 4-layer LSTM + skip connection + averaged weight drop + weight penalty + weight tying [29] | 24M | 58.3 | / |
| LSTM + averaged weight drop + Mixture of Softmax + weight penalty + weight tying [42] | 22M | 56.0 | / |
| NAS + weight tying [45] | 54M | 62.4 | 1e4 CPU days |
| ENAS + weight tying + weight penalty [34] | 24M | 58.6[5] | 0.5 |
| Random-WS + weight tying + weight penalty | 27M | 58.81 | 0.4 |
| DARTS+ weight tying + weight penalty [28] | 23M | 56.1 | 1 |
| NAONet + weight tying + weight penalty | 27M | 56.0 | 300 |
| NAONet-WS + weight tying + weight penalty | 27M | 56.6 | 0.4 |

Table 3: Performance of different models and techniques on PTB dataset. Similar to CIFAR-10 experiment, 'NAONet-WS' represents NAO accompanied with weight sharing,and 'Random-WS' is the corresponding random search baseline.

We leave the model training details of PTB to the Appendix. The encoder in NAO is an LSTM with embedding size $64$ and hidden size $128$. The hidden state of LSTM is further normalized to have unit length. The performance predictor is a two-layer MLP with each layer size as $200, 1$. The decoder is a single layer LSTM with attention mechanism and the hidden size is $128$. The trade-off parameters in Eqn. (1) is $\lambda = 0.8$. The encoder, performance predictor, and decoder are trained using Adam with a learning rate of $0.001$. We perform the optimization process in Alg 1 for two iterations (i.e., $L = 2$). We train the sampled RNN models for shorter time ($600$ epochs) during the training phase of NAO, and afterwards train the best architecture discovered yet for $2000$ epochs for the sake of better result. We use $200$ P100 GPU cards to complete all the process within $1.5$ days. Similar to CIFAR-10, we furthermore explore the possibility of combining weight sharing with NAO and the resulting architecture is denoted as 'NAO-WS'.

We report all the results in Table 3, separated into three blocks, respectively reporting the results of experts designed methods, architectures discovered via previous automatic neural architecture search methods, and our NAO. As can be observed, NAO successfully discovered an architecture that achieves quite competitive perplexity $56.0$, surpassing previous NAS methods and is on par with the best performance from LSTM method with advanced manually designed techniques such as averaged weight drop [29]. Furthermore, NAO combined with weight sharing (i.e., NAO-WS) again demonstrates efficiency to discover competitive architectures (e.g., achieving $56.6$ perplexity via searching in 10 hours).

### 4.4 Transferring the discovered architecture to WikiText-2

We also apply the best discovered RNN architecture on PTB to another language modelling task based on a much larger dataset WikiText-2 (WT2 for short in the following). Table 4 shows the result that NAONet discovered by our method is on par with, or surpasses ENAS and DARTS.

| Models and Techniques | #params | Test Perplexity |
|---|---|---|
| Variational LSTM + weight tying [20] | 28M | 87.0 |
| LSTM + continuos cache pointer [16] | - | 68.9 |
| LSTM [30] | 33 | 66.0 |
| 4-layer LSTM + skip connection + averaged weight drop + weight penalty + weight tying [29] | 24M | 65.9 |
| LSTM + averaged weight drop + Mixture of Softmax + weight penalty + weight tying [42] | 33M | 63.3 |
| ENAS + weight tying + weight penalty [34] (searched on PTB) | 33M | 70.4 |
| DARTS + weight tying + weight penalty (searched on PTB) | 33M | 66.9 |
| NAONet + weight tying + weight penalty (searched on PTB) | 36M | 67.0 |

Table 4: Performance of different models and techniques on WT2 dataset. 'NAONet' represents the best architecture discovered by NAO on PTB.

## 5  Conclusion

We design a new automatic architecture design algorithm named as *neural architecture optimization* (NAO), which performs the optimization within continuous space (using gradient based method) rather than searching discrete decisions. The encoder, performance predictor and decoder together makes it more effective and efficient to discover better architectures and we achieve quite competitive results on both image classification task and language modeling task. For future work, first we would like to try other methods to further improve the performance of the discovered architecture, such as mixture of softmax [42] for language modeling. Second, we would like to apply NAO to discovering better architectures for more applications such as Neural Machine Translation. Third, we plan to design better neural models from the view of teaching and learning to teach [13, 39].

## 6  Acknowledgement

We thank Hieu Pham for the discussion on some details of ENAS implementation, and Hanxiao Liu for the code base of language modeling task in DARTS [28]. We furthermore thank the anonymous reviewers for their constructive comments.

## Footnotes

[3]For ease of introduction, even though some notations have been used before (e.g., $h_t$ in defining RNN search space), they are slightly abused here.

[4] If we have $x' = x$, i.e., the new architecture is exactly the same with the previous architecture, we ignore it and keep increasing the step-size value by $\frac{\eta}{2}$, until we found a different decoded architecture different with $x$.

[5]We adopt the number reported via [28] which is similar to our reproduction.

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
