[Supplementary Material · NAO_supplementary.pdf]

# Supplementary Materials for the Paper "Neural Architecture Optimization"

[1]**Renqian Luo**[†,*] [2]**Fei Tian**,[†] [2]**Tao Qin**, [1]**Enhong Chen**, [2]**Tie-Yan Liu**
[1]University of Science and Technology of China, Hefei, China
[2]Microsoft Research, Beijing, China
[1]lrq@mail.ustc.edu.cn, cheneh@ustc.edu.cn
[2]{fetia, taoqin, tie-yan.liu}@microsoft.com

## 1 Details of Model Training on Image Classification and Language Modelling

CIFAR-10 contains $50k$ and $10k$ images for training and testing. We randomly choose 5000 images within training set as the dev set for measuring the performance of each candidate network in the optimization process of NAO. Standard data pre-processing and augmentation, such as whitening, randomly cropping $32 \times 32$ patches from unsampled images of size $40 \times 40$, and randomly flipping images horizontally are applied to original training set. The CNN models are trained using SGD with momentum set to $0.9$, where the arrange of learning rate follows a single period cosine schedule with $l_{max} = 0.024$ proposed in [1]. For the purpose of regularization, We apply stochastic drop-connect on each path, and an $l_2$ weight decay of $5 \times 10^{-4}$. All the models are trained with batch size 128.

Penn Treebank (PTB) [2] is one of the most widely adopted benchmark dataset for language modeling task. We use the open-source code of ENAS [3] released by the authors and exactly follow their setups. Specifically, we apply variational dropout, $l_2$ regularization with weight decay of $5 \times 10^{-7}$, and tying word embeddings and softmax weights [4]. We train the models using SGD with an initial learning rate of $10.0$, decayed by a factor of $0.9991$ after every epoch staring at epoch 15. To avoid gradient explosion, we clip the norm of gradient with the threshold value $0.25$. For WikiText-2, we use embdedding size 700, weight decay of $5 \times 10^{-7}$ and variational dropout $0.15$. Others unstated are the same as in PTB, such as weight tying.

## 2 Search Space

In searching convolutional cell architectures without weight sharing, following previous works of [5, 6], we adopt 11 possible ops as follow:

- identity
- $1 \times 1$ convolution
- $3 \times 3$ convolution
- $1 \times 3 + 3 \times 1$ convolution
- $1 \times 7 + 7 \times 1$ convolution
- $2 \times 2$ max pooling
- $3 \times 3$ max pooling
- $5 \times 5$ max pooling

---

[*]The work was done when the first author was an intern at Microsoft Research Asia.
[†]The first two authors contribute equally to this work.

- $2 \times 2$ average pooling
- $3 \times 3$ average pooling
- $5 \times 5$ average pooling

When using weight sharing, we use exactly the same 5 operators as [3]:

- identity
- $3 \times 3$ separable convolution
- $5 \times 5$ separable convolution
- $3 \times 3$ average pooling
- $3 \times 3$ max pooling

In searching for recurrent cell architectures, we exactly follow the search space of ENAS [3], where possible activation functions are:

- tanh
- relu
- identity
- sigmoid

## 3 Best Architecture discovered

Here we plot the best architecture of CNN cells discovered by our NAO algorithm in Fig. 1.

Furthermore we plot the best architecture of recurrent cell discovered by our NAO algorithm in Fig. 2.

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

(a) Normal Cell

(b) Reduction Cell

Figure 1: Basic NAONet building blocks. NAONet normal cell (left) and reduction cell (right).

Figure 2: Best RNN Cell discovered by NAO for Penn Treebank.