[Reviews · NeurIPS 2018]

Reviewer 1



[Update after end of author response period] I've left my original (relatively high) score for the submission unchanged. I think the optimization approach proposed in the paper is novel enough to justify the high score, and that the experimental evaluation is sufficient to show that the optimization method is doing something sensible. I also think that the new experiments from the rebuttal will improve the paper, and will help address some of the concerns raised in the original reviews. Comparison with related work: The experiments in this paper show state-of-the-art results. However, in the original submission, it was difficult to tell the extent to which quality and sample efficiency improvements over NASNet/AmoebaNet were coming from the optimization algorithm, as opposed to search space improvements or longer training times. The authors directly address both points in their rebuttal, and I believe the experiments they mention will help explain where the efficiency improvements come from. In particular: (i) shortening training times from 100 epochs to 25, and (ii) increasing the number of operations in the search space from 11 to 19. ENAS Experiments: I believe that including more information about the ENAS experiments mentioned in the rebuttal would significantly improve the final submission. First: it would make the work more relevant to practitioners who do not have access to hundreds of GPUs for experiments. And second: it's currently unclear which search algorithms work well with ENAS's weight sharing scheme, and the fact that NAO can work well in this setting is a nontrivial finding. [Original Review] The authors propose a new method for neural network architecture search. Like "Learning Transferable Architectures for Scalable Image Recognition" (Zoph et al. 2017) and "Regularized Evolution for Image Classifier Architecture Search" (Real et al. 2018), their method runs in a loop where (i) a controller proposes a set of child model architectures, (ii) they measure the accuracy of each child model after training for a fixed number of epochs on CIFAR-10, and (iii) they use the observed model accuracies to generate a new set of improved architectures to evaluate. The main contribution of this paper is a novel method for proposing child model architectures to evaluate. The authors simultaneously train three components: (i) an encoder which maps discrete neural network architectures to a continuous embedding space, (ii) a performance predictor which predicts accuracies for architectures in the continuous embedding space, and (iii) a decoder which maps continuous embeddings back to discrete network architectures. Once these components are trained, it is possible to map a neural network architecture to a continuous embedding, mutate the embedding (via gradient descent) to increase its predicted accuracy, and then map the mutated embedding back to a discrete architecture. This mutation step is somewhat reminiscent of the evolution-based approach proposed by Real et al., except that the mutation function is learned. Originality: The proposed search method (rather than its application to architecture search) is the part of the paper that I find most exciting. The idea of perturbing a continuous-valued embedding to improve a discrete optimization problem is very clever, and I don't think I've seen anything like it elsewhere. It seems like similar techniques should be applicable to other discrete optimization problems, and I'd love to see follow-up work in this direction. Clarity: The paper is well-written and easy to understand. Quality: I'll distinguish between the thoroughness of the experimental evaluation (which is very good) and the usefulness of the method (which is an incremental improvement over previous work). Experimental evaluation: The authors do a great job of setting up their experiments carefully and validating their setup. They show that their quality prediction network gets better at ranking different architectures over the course of a search; ranking seems like the a great metric to consider. They also (critically) include experiments to demonstrate that their architecture mutation step is actually improving the quality of their architectures. Usefulness of the method: * While models evaluated on CIFAR-10 are state-of-the-art, I'm more excited about the general search method than about the specific application to architecture search. * Model accuracies on CIFAR-10 are competitive with or slightly better than previous results by Zoph et al. and Real et al. while using fewer resources. Resource requirements are orders of magnitude greater than weight-sharing methods like Pham et al.'s "Efficient Neural Architecture Search via Parameter Sharing", but the accuracies found in this paper seem slightly better. * In the experiments, the efficiency gains obtained by the search method (evaluating 1,000 architectures instead of 20,000) are partially offset by the increased number of training epochs per candidate architecture (100, compared with 25 in previous work). Looking at Appendix 1.3, it seems possible that changes to the operations in the search space (focusing on operations with small kernels like 2x2 convolutions instead of ops with larger kernels like 5x5 or 7x7 depthwise separable convolutions) may account for some of the efficiency gains. Small question: Appendix 2.3 mentions an activation function called "tanhshi." Is that a typo?

Reviewer 2



======== review after reading the author response Most of my concerns have been reasonably addressed in the author response. The authors also have also provided the requested results of smaller NAO models, which are slightly worse than the SOTA but IMO are sufficient to demonstrate that the proposed method is promising. I've raised my score to the positive side. Btw, I disagree with the authors' point that there's no need to report the performance of random search. It would be informative to report the random search results under the same computation budget as NAO, as it could help verify that the performance gain primarily comes from the algorithm itself rather than some other design choices of the search space/training setup. ======== original review The paper proposed a new architecture search method by encoding the discrete architectures into a continuous vector space, conducting optimization there and then mapping the architecture embeddings back to the discrete space via a decoder. Experiments on both CIFAR and PTB show either improved efficiency or performance over existing methods. Pros: 1. The paper is well written and technically sound. 2. The proposed method is interesting and appears to be very different from existing approaches based on RL and evolution. 3. Results on CIFAR-10 and PTB seem promising. It’s also good to see that the algorithm can handle both convolutional and recurrent cells. Cons: 1. I'm not sure if the authors have fully circumvented the discrete space as they claimed. While the optimization for the architecture embedding is carried out in the continuous space, training the decoder requires learning to represent a discrete set which can be combinatorially large. In fact, one may also view the hidden units in the NAS policy network/controller as continuous representations, and the performance predictor proposed in the paper is therefore analogous to the value network which is more or less standard in RL. 2. The search space used in the paper seems substantially smaller than that of AmoebaNet/NASNet. In particular, there were 19 candidate ops in regularized evolution (Real et al., 2018) whereas the authors only used a small subset of 4 candidate ops. This could potentially make the efficiency comparison unfair. 3. The authors did not report the variance of their models. This could be important as the variance (especially on CIFAR-10) for training the same model could be even larger than the current advantage of NAO over other baselines . Other questions: 1. How did you determine the trade-off hyperparameters in eq. (1) (specified in L199 and L266)? This seems nontrivial as each run of the algorithm would already require 800 GPU days using P100s. 2. How important it is to have the symmetric regularization term (L138-145)? 3. The current best NAO models are quite large. Can you make their sizes comparable with ENAS/PNAS (by using smaller F) and report their performance?

Reviewer 3



>>>>>>>>>>>>>>>>> After the discussion phase Based on the discussion with the area chair assigned to this paper, I would like to comment on the reproducibility of the paper. In fact, I doubt that this paper is easily reproducible (which also had an influence on the score I gave to this paper). I believe (given my own experience with the reproduction of the original NAS-RL by Zoph et al and supported by Footnote 3 of the authors in the paper at hand) that too many factors on different levels are required to reproduce the results: architecture options in the target network to be optimized, hyperparameters to train the target network, architectures of the encoder, performance predictor and decoder, hyperparameters to train the encoder, performance predictor and decoder, meta-hyperparameters of Algorithm 1 and the loss function (\lambda_i). Only missing one little detail can substantially change the results (because DL is very sensitive too many hyperparameter decisions). Because we are all humans, we tend to underestimate certain aspects and forget little, but important details to report in papers. That was again shown in this paper, since I had to ask how the authors optimized the hyperparameters of the target network. (I assume that the authors only forgot this important step and they didn’t leave it out on purpose.) Therefore, to the best of my knowledge, open source code is the best way right now to ensure reproducibility. Another aspect is that I (as a reviewer) are not able to check whether all these important, little details are reported in the paper. If source code would be available (already during the review process), I could be sure that at least the source code covers (nearly) everything. (A docker image would be even better because the dependencies would also be listed, which can also influence the results.) I would like to apologize in advance if I’m too sceptical here; but my experience taught me in the last years that the promise of “We will release all our codes/models after the paper is published.” is not sufficient. A discussion at the AutoML workshop@ICML’18 even supported me in my believe. Therefore, I cannot give credits to the authors because of the promise in the response letter. >>>>>>>>>>>>>>>>> Response to response letter Thank you for your thorough response to the reviews. I really appreciate that you ran more experiments. Unfortunately, because of the space restrictions in the response letter, many details about the experimental setup are missing and I cannot really assess the value of these new results. Nevertheless, the results sound very promising. Finally, I would like to disagree with some of the raised points in the response letter: > We’d like to respectfully point out that it is neither appropriate nor necessary to directly compare NAO with ENAS. Given that we are talking about a NIPS paper, I don't think that I'm asking for too much: a fair comparison between NAO and ENAS. I agree that both approaches are working on different aspects of the underlying problem. Nevertheless, I had the feeling that the paper has not honestly discussed how these approaches could relate and what would happen if NAO and ENAS are combined. > Third, the EE trade-off is naturally achieved via the progress of P: initially P is inaccurate and will lead to worse architectures which plays the effect of exploration. If that would be true, you could also argue that all the effort for exploration-exploitation trade-off in Bayesian Optimization is not necessary, because in BO, we also start with an inaccurate model. However, as shown in the BO literature, the EE-trade-off (achieved by an acquisition function) is important. > we believe there is no need to explicitly report the result of random search Depending on how you design your search space, random search can perform quite well. Therefore, it is an important baseline to really show that you approach can do something better. > he performance growing w.r.t. NAO iteration (in Fig. 2(b)) cannot be achieved via random search How do you know? > Our method is simple grid search I strongly wonder why people are still using grid search these days after there is so much literature on proper HPO. > we only fine-tune My point was that you have not reported HPO in a sufficient manner. For such an expensive approach as NAO, it is crucial how expensive HPO is, how to do HPO in this setting and last but not least, how sensitive your method is wrt to its own hyperparameters. PS: Little details were changed in the original review. >>>>>>>>>>>>>>>>> Original Review: The authors address the challenge of neural architecture search. Since manual design of neural network architectures is a tedious task and requires expert knowledge, automatic neural architecture search was proposed to help new users to easily apply deep learning to their problems. The authors of the paper at hand propose an approach dubbed neural architecture *optimization* (NAO). They call it "optimization" instead of "search" because they optimize the architecture in a continuous space and they do not search in a discrete space. To this end, they propose to (i) learn an encoder to map architectures into a continuous space, (ii) predict the performance of architectures in the continuous space by also using a neural network and (iii) use a decoder from the continuous space into an architecture such that they can evaluate an architecture which was predicted to have a good performance. In the end, they show that this proposed approach finds well performing architectures for CIFAR-10 and Penn Tree bank (PTB). Most previous approaches (which are based on reinforcement learning (RL) and evolutionary strategies (ES)) use indeed a discretized space, but the proposed new approach uses a learned continuous space to search for a good architecture. On the one hand, it is indeed an interesting paper because it proposes a completely new approach. On the other hand, the question arises whether this approach is better performing than already existing approaches or whether we can learn something interesting about the problem. Unfortunately in my opinion, the answer to both questions is no. The authors report that they found well-performing network architectures which perform similarly compared to previous approaches. However, the cost to find these networks is substantially larger than before. For example, ENAS uses 0.45 GPU days to search for a network. The proposed NAO used 800 GPU days on CIFAR-10 and 500 GPU days on PTB. So, it is not feasible in practice. Already the initialization phase of NAO is extremely expensive; the authors evaluated 600 architectures out of the 1000 networks evaluated overall only in the initial phase. Thus, the initialization phase roughly costs 480 GPU days on CIFRA-10; so roughly 1000 times the time for the entire ENAS search. Besides the problem of the experimental results, I would like to raise several other issues: 1. The overall approaches strongly reminds me of Bayesian Optimization (BO). The authors claim that BO is something different but they are doing something very similar: a. based on some performance observations, they build a model b. based on the model, they choose n architectures c. they evaluate these architectures The main difference between BO and NAO is in fact that the authors do not use an acquisition function to trade off exploration and exploitation. However, I wonder why the authors do not have to consider this exploration-exploitation dilemma; maybe because they doing the crazy expensive initialization phase with 600 randomly sampled architectures? 2. The argument of the authors is that optimization in the discrete space is too expensive (although other researchers showed some rather convincing results) and does not scale with the number of architecture hyperparameters. However, I wonder why a mapping into a space with much more dimensions (T x d) should be more efficient. The authors do not make this point very prominent but I guess the reason could be that they can use gradient methods in the continuous space. Nevertheless, a thorough experimental proof of this point is missing. They do not even show that NAO performs better than random search with this tremendous amount of GPU days. 3. Another big issue is the handling of hyperparameter tuning, i.e., the tuning of the hyperparameter of their own NAO and all hyperparameters of the neural networks (e.g., learning rate). The authors wrote: "we have not conducted extensive hyper-parameter exploration as is typically done in previous works". So, the authors have conducted some hyperparameter optimization but they do not tell us how much exactly, how expensive it was and how the hyperparameter tuning influenced the results. Given that they changed the hyperparameters settings between CIFAR-10 and PTB (for example the trade-off parameters of their loss function), I guess that these hyperparameters are indeed crucial and NAO is not robust wrt these. 4. From the perspective of optimization (/search), the number of exemplary problems (here CIFAR-10 and PTB) is far too small. There is only little substantial evidence that the proposed approach performs well and robustly also on future problem instances. Given the advances in other communities (i.e., evaluation on tens of problems), I recommend to evaluate NAO on more datasets to obtain better empirical evidence. I'm well aware that issues 3 and 4 also apply to many other architecture search papers. Nevertheless, these are also major flaws in these papers and it is not an sufficient argument to accept the paper at hand. Further remarks and questions: 1. Introduction * You can also optimize in a discrete space, e.g., the MaxSAT problem is a typical discrete optimization problem. Thus, NAS should be called NAO in the first place. And it is wrong to say that optimization in a discrete space is called "search". * "the continuous representation of an architecture is more compact" I was missing some empirical evidence for this claim. In particular, I wonder why it should be more compact if the number of dimensions in the continuous space is larger than in the discrete space. 2. Related Work * "Using BO, an architecture’s performance is modeled as sample from a Gaussian process (GP)" Although GPs are quite common in the BO community, approaches, such as Hyperopt and SMAC, use different models. 3. Approach * "the outputs of all the nodes that are not used by any other nodes are concatenated to form the final output of the cell". At least in the ENAS paper, the output is not concatenated but averaged * Why do you need all hidden states for the encoding of the continuous? Why is the last state not sufficient? * "where X is a set containing several manually constructed architecture pairs (x 1 , x 2 )" The argument against BO was the manual design of the kernel. However, also NAO requires some manual engineering. * Algorithm 1: How expensive is each step? Or asked differently: How large is the overhead for the embedding, the model training, the surrogate optimization and the decoding? 4. Experiments * "For the purpose of regularization, We apply" --> we * "The detailed results are shown in Table 4.1" -> "The detailed results are shown in Table 1" 5. Conclusion * "makes it more effective and efficient to discover better architectures" Why is it more efficient if NAO requires much more compute resources compared to ENAS? Last but not least, since no link to an open-source implementation is provided (or even promised) and (as the authors experienced themselves) the reproducibility in the field of NAS is rather hard, I would guess that the reproducibility of the paper at hand is also rather poor.